# Mitochondrial DNA and Microsatellite Analyses Showed Panmixia between Temporal Samples in Endangered *Anguilla japonica* in the Pearl River Basin (China)

**DOI:** 10.3390/ani12233380

**Published:** 2022-12-01

**Authors:** Zaixuan Zhong, Huaping Zhu, Jiajia Fan, Dongmei Ma

**Affiliations:** 1Pearl River Fisheries Research Institute, Chinese Academy of Fishery Sciences, Guangzhou 510380, China; 2Key Laboratory of Tropical and Subtropical Fishery Resources Application and Cultivation, Ministry of Agriculture and Rural Affairs, Guangzhou 510380, China; 3Guangdong Provincial Key Laboratory of Aquatic Animal Immunology and Sustainable Aquaculture, Guangzhou 510380, China

**Keywords:** *Anguilla japonica*, panmixia, mitochondrial DNA, microsatellite DNA

## Abstract

**Simple Summary:**

The wild resources of Japanese eel (*Anguilla japonica)* in the Pearl River basin have dramatically declined. Consequently, resource management and conservation of this species are urgently required. By analyzing two mitochondrial fragments (mtDNA) and eight microsatellite markers, we found that nine temporal samples from the Pearl River estuary were not differentiated from each other. In addition, the result of STRUCTURE also supported the conclusion that *A. japonica* was a panmictic population. Therefore, we suggest that the Japanese eel should be managed as a single unit for conservation.

**Abstract:**

The Japanese eel (*Anguilla japonica*) is a commercially important species in East Asia, the abundance of which has rapidly decreased in recent decades. The fishery resource in the Pearl River basin has mainly deteriorated due to overexploitation and habitat degradation. Knowledge on its genetic status is indispensable for resource management. In this study, we explored the temporal genetic structure of *A. japonica* on the basis of the concatenated sequences of two mitochondrial fragments (mtDNA) and eight microsatellite markers. A total of nine temporal samples (N = 127) were collected during 2019 and 2021 from Jiangmen City, China, which is located in the Pearl River estuary. mtDNA sequence analysis showed a high level of haplotype diversity, and yielded 124 haplotypes with ranging from 9 to 19 in temporal samples. All microsatellite loci were polymorphic among each of the nine temporal samples, with 150 alleles identified across all samples. Pairwise F_ST_ values were low and nonsignificant according to both mtDNA and microsatellite markers. STRUCTURE analysis showed that all temporal samples were not clearly differentiated from each other. The yielded outcomes supported a panmictic pattern in different temporal *A. japonica* samples. Therefore, our results call for the management of *A. japonica* as a single unit and joint conservation strategy of the species, since overexploitation in any region will decrease its global resource.

## 1. Introduction

Japanese eel (*Anguilla japonica*), a catadromous species, is widely distributed in China (China mainland and Taiwan Island), Japan, and Korea. It has a particular and complex life cycle, spawning to the west of the Mariana Ridge [1] and swimming toward nearby estuaries and rivers for further growth [2,3]. Due to its good flavor and high nutrition, *A. japonica* has become economically important in terms of market demand [4]. However, the technology of artificial breeding has not been solved yet, resulting in intensive capture of glass eel for commercial aquaculture [5]. Inevitably, the biomass of eels has drastically decreased in the last few decades [6]. Moreover, factors including pollution, climate change, and habitat losses also cause a decrease in the *A. japonica* population [7,8]. *A. japonica* was classified as an endangered species in Japan by the Ministry of the Environment (2013, https://www.env.go.jp/press/15619.html (accessed on 10 May 2021)) and was placed on the Red List of Threatened Species in 2018 (https://www.iucnredlist.org/species/166184/176493270 (accessed on 10 May 2021)). Therefore, assessing the genetic structure and diversity of this endangered species is essential to provide scientific management and protection of its population resources.

However, controversy still exists as to whether *A. japonica* is made up of single panmixia or multiple geographically based populations [7,9,10]. Tseng et al. [11] used the mitochondrial DNA D-loop to analyze temporal genetic variations of *A. japonica* collected from Taiwan Island in 1989–2008 and found that most of the pairwise F_ST_ appeared statistically not significant between different populations of *A. japonica*. In addition, no temporal isolation was identified using restriction site-associated DNA sequencing (RAD-seq) technology across five “cohort” samples collected yearly from 2005 to 2009 in the Yangtze River Estuary [10]. Yu et al. [7] developed twenty-four polymorphic gene-associated microsatellite markers to delineate the genetic structure of eleven samples from China (nine from China’s mainland and two from the Taiwan region) and two samples from Japan, which further confirmed panmixia in *A. japonica*. However, there exist conflicting hypotheses about the population genetic structure of *A. japonica*. For example, *A. japonica* on the western Pacific fringe exhibited clear geographic clines using the IDHP and PGDH allozyme loci [12]. Research based on eight microsatellite loci revealed that the *A. japonica* collected from seven locations could be divided into two major groups including a low-latitude group and a high-latitude group [13].

The Pearl River estuary is a key area in southern China for concentrated fishing of juvenal eel, owing to a richness in bait and a suitable ecological environment [14]. A preliminary investigation of *A. japonica* in the Pearl River basin during 2015 and 2017 uncovered a serious scarcity of the germplasm resource [15]. However, the genetic diversity and structure of *A. japonica* in the Pearl River basin remain unclear, which hinders us from establishing protection strategies.

Therefore, in this study, we collected nine *A. japonica* temporal samples in the Pearl River estuary during 2019 and 2021 and evaluated the genetic diversity using both mitochondrial loci and microsatellite DNA markers. We aimed to (i) verify whether there is a temporal genetic structure of *A. japonica* or not, and (ii) provide helpful suggestion of resource protection and sustainable utilization of *A. japonica*. 

## 2. Materials and Methods

### 2.1. Sample Collection and DNA Extraction

A total of 127 individuals of *A. japonica* were collected in nine temporal samples from Jiangmen City, Guangdong Province (Figure 1) from January 2019 to September 2021 (Table 1). All samples were silver eel collected with a net. The caudal fins of eels were washed several times with nuclease-free water and preserved in 99.9% ethanol until DNA extraction. Genomic DNA was extracted using Axyprep Multisource Genomic Miniprep DNA Kit (Axygen, Hangzhou, China) according to the manufacturer’s protocol.

### 2.2. Mitochondrial DNA Sequencing

The primers for the Cyt *b* gene (forward primer: 5′ CCTCCTTCTTCTTTATCTGCCTC 3′, reverse primer: 5′ GTTTTCTAGTCAACCTGCTAATGG 3′) and D-loop region (forward primer: 5′ GCATCGGTTTTGTAATCCG 3′, reverse primer: 5′ GGGGATATAGGGCATTCTCA 3′) were designed on the basis of the mitochondrial genome of *A. japonica* (GenBank: AB038556.2). Polymerase chain reaction (PCR) was performed in the AppliedBiosystem Veriti96 system (ThermoFisher, Waltham, MA, USA) with a total reaction volume of 25 μL composed of 12.5 μL PCR Master Mix (GeneTech, Shanghai, China), 9.5 μL water, 1 μL of each primer and 1 μL genomic DNA. The PCR conditions were as follows: an initial denaturation at 94 °C for 3 min, followed by 30 cycles of denaturation at 94 °C for 1 min, annealing at 60 °C for 1 min, extension at 72 °C for 1 min, and final extension at 72 °C for 5 min. The amplified product was evaluated for quality using 1% agarose gels and sequenced on 3730 xl DNA Analyzer (Applied Biosystems, Bedford, MA, USA).

### 2.3. Microsatellite Genotyping

Eight microsatellite loci including two (GA)_n_ and six (GT) _n_ loci were selected from GenBank (Appendix A). These simple sequence repeat markers (nSSRs) were amplified as described previously [16]. PCR was performed in a 10 μL reaction volume containing 5 μL PCR Master Mix (GeneTech, Shanghai, China), 3 μL water, 0.5 μL of each primer and 1 μL genomic DNA. The PCR condition consisted of an initial denaturation at 94 °C for 3 min followed by 30 cycles with denaturation at 94 °C for 30 s, annealing at 56 to 60 °C for 30 s, extension at 72 °C for 30 s, and a final extension at 72 °C for 10 min. The 2 µL PCR products were taken for agarose gel electrophoresis detection (1% concentration) to determine the specificity and to judge the amplification efficiency of each SSR primer. According to the concentration requirements of each sample, fluorescent PCR products were diluted to a uniform concentration. All amplification products were genotyped through fluorescence capillary electrophoresis (ABI 3730xl, Applied Biosystems, Bedford, MA, USA).

### 2.4. Data Analysis for mtDNA

Amplified sequences were assembled using SEQMAN [17] and were aligned and trimmed using MEGA X [18]. Consensus sequences were formed by concatenating the Cyt *b* sequences and D-loop sequences for subsequent analyses. DNASP 6.0 [19] was implemented to estimate the genetic diversity indices, including the number of haplotypes (H), haplotype diversity and nucleotide diversity (π). Pairwise genetic differentiation was assessed from genetic differentiation indices (F_ST_) using ARLEQUIN v3.5 [20]. To estimate the genetic variation partitioning among temporal samples, analysis of molecular variance (AMOVA) was implemented in ARLEQUIN v3.5. All calculations performed in ARLEQUIN were based on a nonparametric permutation procedure (1000 permutations/analysis).

To determine the demographic history of *A. japonica* samples, neutrality tests and Bayesian skyline analyses were carried out. Firstly, Fu’s *F_S_* [21] and Tajima’s *D* [22] tests were implemented in ARLEQUIN 3.5 to check the demographic expansion. Secondly, a Bayesian skyline plot analysis with a nonparametric estimation was conducted to infer the fluctuation in the number of effective populations using the Bayesian Markov chain Monte Carlo (MCMC) method in BEAST v1.8.1 [23]. According to a previous study, the evolutional rate of D-loop region in teleosts is 3.60% per million years [24]. The average net K2P distance for the consensus sequences vs. D-loop region alone was 0.708. Therefore, the estimated mean substitution rate for the consensus sequences (2.55%) was obtained by multiplying the D-loop region rate by the K2P ratio for D-loop region alone (0.708). The optimal nucleotide substitution model (GTR + I + G) determined using MrModeltest v2.3 [25] was employed in Bayesian skyline analysis. A strict clock model was set as prior, 10^9^ generations were run with sampling every 1000 iterations. The convergence and diagram of Bayesian skyline analysis were checked in TRACER 1.5 [26]. Lastly, mismatch distribution analysis was performed to detect the demographic history and expansion using DNASP 6.0 and ARLEQUIN v3.5.

### 2.5. Data Analysis for nSSR

The program GeneMarker 3.0.0 [27] was used to read and export the genotype data. To examine the presence of large allele dropout, shuttering, and null alleles, genotype data were analyzed with MICRO-CHECKER [28]. Genetic diversity was assessed by calculating the allele frequencies at each locus for each population, the average number of alleles per population, observed (A) and effective numbers of alleles (Ae), observed (Ho) and expected heterozygosity value (He), Shannon information index (I), and heterozygote deficit (F_IS_) per locus across populations and markers using GenAIEx 6.5 [29]. Deviations from Hardy–Weinberg expectations (HWEs) for all loci were tested using ARLEQUIN v3.5 [20]. Allelic richness (Ar) was evaluated according to the rarefaction method using FSTAT 2.9.4 (https://www2.unil.ch/popgen/softwares/fstat.htm (accessed on 2 June 2022)).

Genetic differentiation between populations was estimated from the fixation index (F_ST_) using ARLEQUIN. The *p* value of pairwise F_ST_ was adjusted using false discovery rate (FDR) correction [30]. To partition total genetic variation among populations, analysis of molecular variance (AMOVA) was implemented in ARLEQUIN. Furthermore, Bayesian analysis under an admixture model was carried out in STRUCTURE v2.3.4 to assess the genetic population structure [31]. The simulation was iterated ten times for each K-value (2–8). The program was run with a 300,000 burn-in period and 500,000 Markov chain Monte Carlo (MCMC) repetitions. CLUMPP [32] was used to align the replicates of each K-value to facilitate the explanation of clustering results, which was graphically displayed by DISTRUCT [33]. The optimal number of groups (K) was determined on the basis of the value of Δ*K* and the log-likelihood method, which was calculated and plotted using Structure harvester [34,35].

## 3. Results

### 3.1. Genetic Variation and Differentiation Based on Mitochondrial DNA

In this study, we obtained 784 bp of Cyt *b* gene and 1150 bp of D-loop for 125 individuals after alignment and trimming. Concatenated sequence analysis yielded 124 haplotypes in total (Table 1). The number of haplotypes ranged from 9 (May20) to 19 (Apr19). The overall haplotype diversity and nucleotide diversity (π) were 0.997 ± 0.001 and 0.005 ± 0.0002, respectively (Table 1). Hd was high in all temporal samples, with the lowest in Jan20 (0.994 ± 0.019) and the highest in May20 (1.000 ± 0.052). As for π, no big difference was found within each temporal sample, ranging from 0.005 ± 0.0004 to 0.006 ± 0.0007.

In the pairwise F_ST_ comparison of different temporal samples, F_ST_ (below diagonal) was low and nonsignificant (Table 2), ranging from −0.032 to 0.013. According to the AMOVA results, the majority (98.88%) of the genetic variation was attributed to variation within temporal samples (Table 3). The genetic differentiation among temporal samples was low (0.011) and not significant (*p* = 0.69).

### 3.2. Microsatellite Marker Variation and Genetic Diversity

All loci were polymorphic among the nine temporal samples. A total of 150 alleles were identified across all samples, ranging from 4 (AJ297601) to 31 (AJ297600), with an average of 18.75 alleles per locus (Table 4). The observed number of alleles per locus (A) ranged from 3 (AJ297601) to 15 (AM062761), while the effective number of alleles (Ae) ranged from 1.75 (AJ297601 to 11.74 (AM062761) with a mean number of 6.94 ± 0.98. Allele richness (Ar) estimates varied from 3.04 to 10.70 with a mean value of 8.50 (Table 4). 

As for the genetic diversity analysis of different temporal samples, the most polymorphic temporal sample was Nov19 with 89 alleles (Table 5). Ae varied from 5.27 (Mar20) to 7.49 (Apr21), with an average of 6.62 ± 0.24 (Table 5). According to the value of Shannon’s information index (I), all microsatellite makers were polymorphic. Moreover, the average value of observed heterozygosity across the loci and temporal samples was 0.70, suggesting a high level of genetic diversity within *A. japonica*.

### 3.3. Genetic Differentiation and Structure Based on Microsatellite Markers

The F_ST_ comparison was calculated to determine the level of heterogeneity across different Japanese eel temporal samples. The results showed that a deficit of heterozygotes (F_IS_) of 9% existed in the studied loci. Across the eight markers, relatively low differentiation was found within nine temporal samples (Table 4). According to the pairwise comparison of different samples based on microsatellite markers, the F_ST_ value ranged from −0.001 to 0.029, none of which was significant after false discovery rate (FDR) correction. The highest differentiation was observed between Mar20 and May20 (F_ST_ = 0.029). AMOVA revealed that the majority (98.63%) of the genetic variation originated from variation within temporal samples, which was consistent with the result of mitochondrial DNA.

Replicate STRUCTURE runs were performed to determine the number of genetic clusters, which overwhelmingly suggested K  =  3 as the most likely scenario according to ΔK. Nevertheless, the K = 3 plot showed that all groups were not clearly differentiated from each other (Figure 2).

### 3.4. Demographic History

Since the values of Fu’s *F_S_* (−24.38) and Tajima’s *D* (−2.29) tests were significantly negative (*p* < 0.001), it is reasonable to suggest that population expansion exists in *A. japonica* temporal samples in the Pearl River. Mismatch distribution analysis revealed a unimodal distribution, indicating that the populations experienced a rapid expansion. The Bayesian skyline plot showed that *A. japonica* temporal samples underwent a rapid increase approximately 0.05 million years ago (Mya) (Figure 3). 

## 4. Discussion

### 4.1. Genetic Diversity

In this study, analysis of mtDNA indicated high haplotype diversity across all temporal samples (Table 1), which concurs with previous studies on *A. japonica* (Hd = 0.916), *A. anguilla* (Hd = 0.997) and *A. bicolor pacifica* (Hd = 0.916) [11,36,37]. According to the microsatellite diversity and differentiation, *A. japonica* samples exhibited sufficient diversity, which was evidenced by the number of alleles and heterozygosity (Table 5). The mean observed number of alleles (A) across temporal samples was greater than nine, similar to that in a previous study [7]. Ho ranged from 0.63 to 0.75, which is consistent with the other two studies (Xinhui City, China, Ho = 0.760, *n* = 25; Zhongshan City, China, Ho = 0.690, *n* = 35) in the Pearl River Basin [7,38].

### 4.2. Evidence of Panmixia 

*A. japonica* has long been considered as a panmictic population due to its catadromous character. The sexually mature stocks migrate to the spawning area, which is presumed to be in the western Mariana Islands near 14°–16° N, 142°–143° E [1]. The leptocephalus larvae disperse from their spawning site via the North Equatorial Current (NEC), followed by the Kuroshio Current (KC), reaching their habitats randomly [39]. To illustrate the population structure of *A. japonica*, various genetic markers such as mitochondrial DNA, microsatellite DNA markers, and single-nucleotide polymorphisms (SNPs) have been exploited [10,40,41]. However, there is still no consensus on whether *A. japonica* is a panmictic population [7,12,13].

In this study, we corroborated that *A. japonica* existed as a panmictic population with no temporal isolation in the Pearl River basin. On one hand, none of the pairwise F_ST_ values between samples was statistically significant according to both mitochondrial DNA and microsatellite DNA markers (Table 2), indicating no temporal differentiation. On the other hand, STRUCTURE analysis revealed that all temporal samples showed similar patterns when K = 3, showing no temporal genetic structure (Figure 2), which is similar to results found by Yu et al. [7]. Two of the most comprehensive studies so far on *A. japonica* also confirmed panmixia of this species at spatial and temporal scales using microsatellite markers and SNPs [10,16]. Moreover, a study collecting glass eel yearly in northern Taiwan Island from 1986 to 2007 found that genetic differentiation was temporally stable at a single location [41]. Meanwhile, conclusions of spatial or temporal variation have been questioned considering that loci deviating from the Hardy–Weinberg equilibrium were used or that random genetic drift across generations existed [16,42]. Additionally, the panmictic phenomenon was also verified in *Anguilla anguilla* and *Anguilla bicolor pacifica*, indicating that panmixia is a common feature of eel species [36,37]. In addition, other migratory species such as Hilsa shad (*Tenualosa ilisha*), Delta Smelt (*Hypomesus transpacificus*) and Australian yellowfin bream (*Acanthopagrus australis*) also exhibited a panmictic character [43,44,45]. 

### 4.3. Demographic Expansion

Neutrality tests displayed significant negative values for both Fu’s *F_S_* and Tajima’s *D* tests, indicating the population expansion of *A. japonica* in the Pearl River basin. Furthermore, the Bayesian skyline plot also demonstrated that population expansion (Figure 3) occurred about 0.05 Mya, which was around the Last Pleistocene era (0.126–0.018 Mya) in southern China. Southern China experienced multiple glacial periods during the Middle Pleistocene era (0.786–0.126 Mya) and entered an interglacial period during the Last Pleistocene era [46]. A warmer interglacial era caused an increase in sea level, changes in current patterns and upwelling intensities, and richness of nutrients, thus facilitating a population size expansion [6]. Moreover, the phenomenon of population expansion around the Pleistocene era was also reported in other eels such as *A. pacifica*, *A. marmorata* and *A. obscura* [35,47,48].

### 4.4. Implications for Conservation

The natural stock of *A. japonica* has rapidly decreased in the past few decades, resulting in a loss of this commercial resource. The average catch of glass eels in East Asia has dropped to <10% of the resources in the 1970s [49]. Factors including overfishing, habitat loss or environmental degradation, construction of upstream dams, water pollution, climate change, and global warming are the main reasons for the decline in *A. japonica* [3,8]. In this study, we confirmed the panmixia of *A. japonica* in the Pearl River basin, which we suggest to be managed as a single unit. Therefore, it is recommended to coordinate protection efforts at the transnational level, since deterioration in any region will decrease recruitment across its global distribution range. First, strengthening the management of glass eel overfishing is a crucial measure to protect resources. Second, conservation strategies for habitat restoration including the establishment of protected areas, checking of dam removal, eel ladder setting, and sewage treatment are urgently needed to protect threatened eel species and effectively recover the carrying capacity of the habitats [8].

## 5. Conclusions

For the first time, our study combined mitochondrial loci and microsatellite DNA markers to confirm panmixia on a temporal scale of *A. japonica* in southern China. No genetic differentiation was identified within *A. japonica* temporal samples, suggesting that it should be managed as a single unit. Moreover, demographic expansion of *A. japonica* was detected during the Last Pleistocene era. Although our study on the genetic structure in *A. japonica* is far from complete due to discontinuous sampling, the results are suggestive for sustainable management.

## Figures and Tables

**Figure 1 animals-12-03380-f001:**
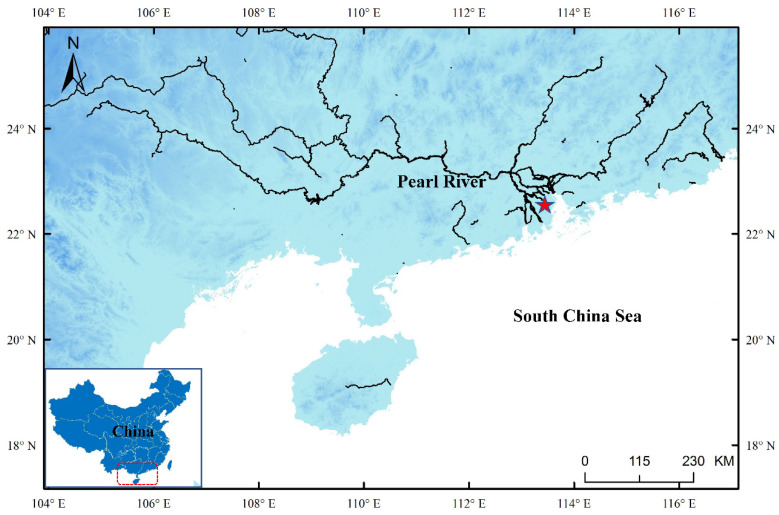
Sampling site of *Anguilla japonica* (red star). This map was generated using ArcGis software.

**Figure 2 animals-12-03380-f002:**
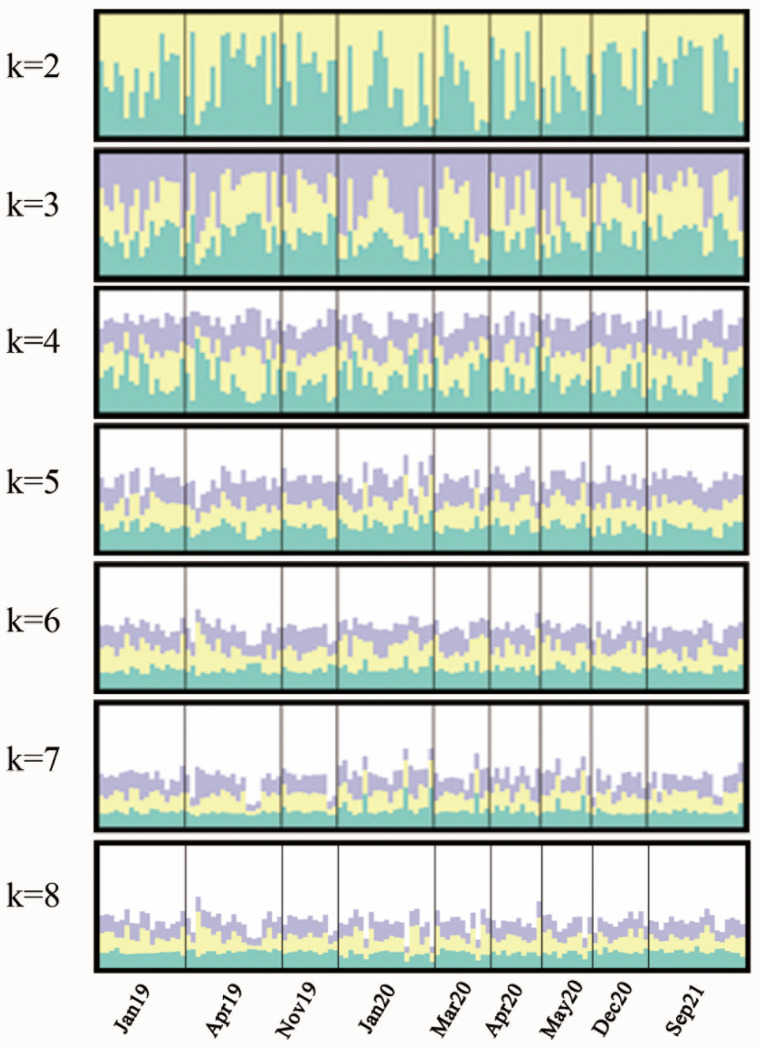
Bayesian clustering of *Anguilla japonica* temporal samples under assumption of K = 2–8, using STRUCTURE v2.3.4. Graphics were generated with DISTRUCT and CLUMPP. Jan19 = January 2019, Apr19 = April 2019, Nov19 = November 2019, Jan20 = January 2020, Mar20 = March 2020, Apr20 = April 2020, May20 = May 2020, Dec20 = December 2020, Sep21 = Sep 2021.

**Figure 3 animals-12-03380-f003:**
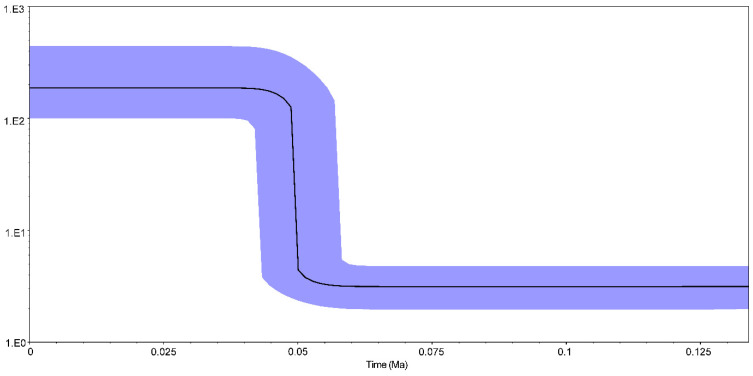
Bayesian skyline plots of *Anguilla japonica* based on a 2.55% evolutionary rate.

**Table 1 animals-12-03380-t001:** Genetic diversity indices of nine temporal samples based on mitochondrial DNA.

Temporal Sample	N	H	Hd	π
Jan19	17	17	1 ± 0.020	0.005 ± 0.0004
Apr19	19	19	1 ± 0.017	0.005 ± 0.0005
Nov19	11	11	1 ± 0.039	0.005 ± 0.0004
Jan20	19	18	0.994 ± 0.019	0.006 ± 0.0005
Mar20	11	11	1 ± 0.039	0.006 ± 0.0007
Apr20	10	10	1 ± 0.045	0.005 ± 0.0004
May20	10	9	1 ± 0.052	0.006 ± 0.0007
Dec20	11	11	1 ± 0.039	0.005 ± 0.0005
Sep21	19	18	1 ± 0.019	0.005 ± 0.0006
Overall	127	124	0.997 ± 0.001	0.005 ± 0.0002

N = Sample size; H = Number of haplotypes; Hd = Haplotype diversity; π = Nucleotide diversity; Jan19 = January 2019; Apr19 = April 2019; Nov19 = November 2019; Jan20 = January 2020; Mar20 = March 2020; Apr20 = April 2020; May20 = May 2020; Dec20 = December 2020; Sep21 = Sep 2021.

**Table 2 animals-12-03380-t002:** Pairwise population matrix of F_ST_ values based on mitochondrial DNA (below diagonal) and microsatellite (above diagonal). None of the pairwise F_ST_ was significant.

Temporal Sample	Jan19	Apr19	Nov19	Jan20	Mar20	Apr20	May20	Dec20	Sep21
Jan19	0	0.010	0.005	0.018	0.017	0.011	0.008	0.011	0.009
Apr19	0.013	0	0.013	0.025	0.021	0.022	0.017	0.014	0.009
Nov19	−0.019	−0.013	0	0.014	0.028	−0.001	−0.001	0.0004	0.003
Jan20	−0.016	0.005	−0.014	0	0.023	0.015	0.0002	0.022	0.022
Mar20	−0.001	0.003	−0.021	0.003	0	0.018	0.029	0.016	0.0002
Apr20	−0.011	−0.007	−0.018	−0.0101	−0.009	0	0.010	0.023	0.019
May20	0.004	−0.018	−0.029	−0.005	0.001	−0.010	0	0.014	0.010
Dec20	−0.021	−0.0004	−0.040	−0.028	−0.017	−0.026	−0.032	0	0.007
Sep21	−0.015	−0.014	−0.017	−0.013	−0.002	−0.023	−0.009	−0.022	0

**Table 3 animals-12-03380-t003:** Analysis of Molecular Variance (AMOVA) results for Japanese eel.

Source of Variation	d.f.	Sum of Squares	Variance Components	Percentage of Variation	Fixation Index
Among temporal samples	8	34.49	−0.06 Va	1.12	0.011
Within temporal samples	116	589.75	5.08 Vb	98.88	
Total	124	624.24	5.03		

d.f. = degree of freedom.

**Table 4 animals-12-03380-t004:** Mean diversity indices and F-statistic (F_IS_, F_IT_, F_ST_) values across eight microsatellite markers.

Locus	Total Allele	A	Ae	I	Ho	He	uHe	F	Ar	Fis	Fit	F_ST_
AM062762	17	9	5.97	1.95	0.75	0.83	0.86	0.09	8.21	0.09	0.13	0.05
AJ297601	4	3	1.75	0.77	0.49	0.42	0.43	−0.17	3.04	−0.17	−0.11	0.05
AJ297602	16	10	7.03	2.08	0.69	0.85	0.89	0.20	8.72	0.20	0.23	0.05
AB051094	13	9	5.40	1.88	0.80	0.81	0.84	0.01	7.78	0.01	0.05	0.04
AM062761	29	15	11.14	2.52	0.70	0.91	0.94	0.23	10.70	0.23	0.27	0.05
AJ297603	19	9	5.97	1.95	0.72	0.82	0.86	0.13	7.94	0.13	0.18	0.06
AJ297600	31	12	8.25	2.27	0.70	0.88	0.91	0.20	10.48	0.20	0.24	0.05
AB051084	21	13	10.02	2.41	0.85	0.90	0.93	0.06	11.17	0.06	0.10	0.04
Mean	18.75	10.01	6.94	1.98	0.70	0.80	0.83	0.09	8.50	0.09	0.13	0.05
SE	2.88	1.12	0.98	0.18	0.04	0.05	0.06	0.04	0.86	0.04	0.04	0.00

A = Number of alleles; Ae = Number of effective alleles; I = Shannon’s information index; Ho = Observed heterozygosity; He = Expected heterozygosity; uHe = Unbiased expected heterozygosity; F = Heterozygote deficiency (inbreeding coefficient); Ar = Allelic richness.

**Table 5 animals-12-03380-t005:** Genetic diversity indices of eight microsatellite markers in nine temporal samples.

Temporal Sample	Total Alleles	A	Ae	I	Ho	He	uHe	F	Ar
Jan19	85	10.63	7.35	2.04	0.72	0.81	0.84	0.09	8.78
Apr19	84	10.50	6.67	1.98	0.69	0.80	0.82	0.11	8.27
Jun19	72	9.00	6.37	1.90	0.74	0.80	0.83	0.05	8.64
Nov19	89	11.13	7.08	2.02	0.69	0.79	0.82	0.11	8.58
Jan20	69	8.63	6.32	1.87	0.63	0.77	0.81	0.16	8.34
Mar20	62	7.75	5.27	1.77	0.71	0.78	0.82	0.06	7.75
Apr20	72	9.00	7.09	1.98	0.75	0.81	0.86	0.06	8.41
May20	69	8.75	5.98	1.87	0.72	0.79	0.83	0.10	8.67
Apr21	86	10.75	7.49	2.07	0.69	0.82	0.85	0.15	9.09
Mean	76.44	9.57	6.62	1.94	0.70	0.80	0.83	0.10	8.50
SE	3.20	0.40	0.24	0.03	0.01	0.01	0.01	0.01	0.13

## Data Availability

All data sets presented in this study have been submitted to GenBank 315 (OP615404–OP615528, OP615529–OP615653).

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
