# Peer review of "Mitochondrial DNA and Microsatellite Analyses Showed Panmixia between Temporal Samples in Endangered Anguilla japonica in the Pearl River Basin (China)"

_animals, 2022, doi:10.3390/ani12233380_

Round 1
Reviewer 1 Report
I had the pleasure to review the manuscript entitled “Mitochondrial DNA and microsatellite analyses confirm panmixia at temporal scale in endangered Anguilla japonica in the Pearl River basin” by authors Zhong et al. While I really enjoyed reading the manuscript, I found several major issues that need to be corrected by authors on the current version from the manuscript. I strongly suggest addressing all the concerns to improve the quality of the manuscript.
Major concerns:
1.- What are bolded values from Table 2? Are these significant FST values? If so, this is evidence of population differentiation and not panmixia, as authors suggest. Tables 3 support this, since the results from the AMOVA suggest a significant differentiation among populations and thus, these result change the title of the manuscript and the main conclusions.
2.- I am concerned about the Hd values from Table 1. Haplotype diversity (d) can range from 0 – 1, but in table 1, authors present values 1 +-0.019- 0.039, implying that Hd can reach values > 1. I am afraid authors rounded up values near to 1. I strongly suggest checking these values.
3.- Also, such high values as the ones presented by authors should also be considered cautiously, since they can be a sign of misidentification of sequencing results. From the methodology, it is not clear to me whether authors sequenced both, forward and reverse fragments of their sequences, neither what criterion authors used to concatenate the D-loop and Cytb genes.
4.- What parameters were used to run a Bayesian skyline plot? There are several errors in the methodology of the Bayesian skyline plot. For instance, authors state that the Bayesian Skyline plot was generated in TRACER, while this is not possible at all.
5.- Did authors check whether all microsatellite markers were neutral? I am assuming that authors did it but forgot to mention it and this should always be included. Otherwise, this test must be assessed to check whether all markers are useful or not.
Minor concerns:
1.- Figure 1 must be edited. It is a very difficult for readers outside of China to locate the sampling region with the map from figure 1. I suggest making changes to the map to include a small map from China in the corner to highlight where the sampling point was located.
Reviewer 2 Report
1.Since this species is an endangered species, the manuscript should provide the fishing permit for the samples.
2.Since there is no genetic differentiation on a larger scale, it is not surprisingly that the samples collected at a fixed place showed no genetic differentiation. The authors should think about the scientific problem of this study.
3.The results of the two fragments of mtdna should be presented separately because their evolutionary rate are different.
Reviewer 3 Report
The authors present some interesting findings involving genetic characterization of the Japanese eel populations. Certainly, the described species is worth an analysis done by the authors, especially due to its position on the commercial market as well as the problem of overfishing. I rate the work highly and I have only a few minor comments:
1. (Figure 1) - The red star is so tiny that it is almost invisible. Please correct this marking.
2. (2.2. Mitochondrial DNA Sequencing) - Please add information about the mixture for the PCR reaction.
3. (2.3. Microsatellite Genotyping) - Same as above.
4. (4.2. Confirmed Panmixia) - I think that this section lacks examples of other fish species in the context of panmixia and confronting them with A. japonica described here. One last sentence, however, is not enough, especially as it only applies to other species of eels.
Reviewer 4 Report
A pdf file with the review was attached.

Round 2
Reviewer 1 Report
Dear authors,
I have reviewed the revised version of your manuscript and I still think that you need to reevaluate the main result and title of your article. Your FST results revealed six low but significant differences between samples. This result contradicts your main conclusion about panmixia, since there is evidence of population differentiation through time. Previous authors have found similar low FST but significant values for Anguilla species (e.g. Minegishi, Y., Gagnaire, P. A., Aoyama, J., Bosc, P., Feunteun, E., Tsukamoto, K., & Berrebi, P. (2012). Present and past genetic connectivity of the Indo‐Pacific tropical eel Anguilla bicolor. Journal of Biogeography, 39(2), 408-420), and thus you should revise your main conclusions since panmixia is not supported by your results.
Also, you should review your figure and table captions, include the species name and location in all of them and also correct minor errors as on line 195 “cross 8microsatellite markers” where it should say “8 microsatellite markers”.
Reviewer 2 Report
The gene flow should be caculated for different populations by some software such as LAMARC. It is confusing that there is Nm in Table 4.
Reviewer 4 Report
Journal: Animals (ISSN 2076-2615)
Manuscript ID: animals-2008757
Type: Article
Title: Mitochondrial DNA and microsatellite analyses confirm panmixia at temporal scale in endangered Anguilla japonica in the Pearl River basin
Authors: Zaixuan Zhong, Huaping Zhu, Jiajia Fan, Dongmei Ma *
Section: Ecology and Conservation
In this new version of the manuscript, the authors have implemented all the suggested changes recommended in the revision, adequately arguing the issues raised. From my point of view, the quality of the manuscript has improved adequately. Nevertheless, I observed some typos that must be corrected.
CONTENT ISSUES
Results
I strongly recommend using at most three decimal places for the values in Table 1 and Table 2. This will help the readability. Please modify the values in the main text consistently.
Line 209: Please change "populations" to "temporal samples".
Lines 213-214: "none of which was significant after false discovery rate (FDR) correction (Table 2, above diagonal)". But according to the header of Table 2 above the diagonal are the Fst values obtained from microsatellite data. This can be confusing, please review it. If my interpretation is correct, I would delete "(Table 2, above diagonal)" in the main text (lines 213-214).
Discussion
Line 262: Please change "demonstrating" to "showing".
Supplementary material
Table S1 (header): I recommend modifying the header. Something like this:
"Table S1. Characteristics of 8 microsatellite loci in Anguilla japonica".
Table S1: I would change the first column name. From "Locus" to "Locus (GenBank Accession)".
FORMATTING ISSUES
Results
Line 183: Please change "among temporal sample" to "among temporal samples", plural.
Line 186 (Table 2 header): Please change "is" to "was".
References
Line 436: Please, remove the last dot "."
Round 3
Reviewer 1 Report
Dear authors,
I have reviewed the new version of the manuscript and I am pleased to say that all my concerns have been responded and addressed.
Author Response
Dear reviewer
Thanks very much for your advice.
Best regards